# Occupational physical activity as a determinant of daytime activity patterns and pregnancy and infant health

Alexis Thrower[1]*, Tyler Quinn[2], Melissa Jones[3], Kara M. Whitaker[4], Bethany Barone Gibbs[2]

**1** Department of Pathophysiology, Rehabilitation, and Performance, West Virginia University, Morgantown, West Virginia, United States of America, **2** Department of Epidemiology and Biostatistics, West Virginia University, Morgantown, West Virginia, United States of America, **3** Department of Human Movement Science, Oakland University, Rochester, Michigan, United States of America, **4** Department of Health and Human Physiology, Department of Epidemiology, University of Iowa, Iowa City, Iowa, United States of America

* ant00017@mix.wvu.edu

**Data Availability Statement:** All relevant data are within the manuscript and its Supporting information files.

## Abstract

Though physical activity (PA) is recommended during pregnancy, it remains unclear how occupational physical activity (OPA) and sedentary behavior (SB) contribute to activity patterns and health during pregnancy. The purpose of this secondary analysis was to determine if OPA pattern is a determinant of all-day PA and evaluate associations with pregnancy/infant health outcomes. Data was from two prospective cohorts with study visits each trimester: MoM Health (Pittsburgh, PA; n = 120) and PRAMS (Iowa City, Iowa; n = 20). Using employment status/job hours (self-reported in demographic questionnaires) and OPA from the Pregnancy Physical Activity Questionnaire, latent class analysis identified three groups: sitting (n = 61), part-time mixed (n = 9), and active (n = 29). A fourth group included non-working participants (n = 32). Device-based PA (ActiGraph GT3X), SB (activPAL3 micro), and blood pressure were measured each trimester. Glucose screening test, gestational age, gestational weight gain, adverse pregnancy outcomes (APOs: gestational hypertension, preeclampsia, eclampsia, gestational diabetes, intrauterine growth restriction, and preterm birth), and infant outcomes (length, weight, and sex) were abstracted from medical records. Associations between groups with APOs and pregnancy/infant health were calculated using linear/logistic regression with adjustment for age, pre-pregnancy BMI, education, and race. Self-reported participant characteristics were similar across groups, except education which was higher in the sitting versus other groups. All-day device-based PA differed across groups; for example, the sitting group had the highest SB across trimester (all p<0.01) while the active group had the highest steps per day across trimesters (all p<0.01). Pregnancy/infant health did not differ between groups (all p>0.09). Compared to the non-working group, the risk of any APO was non-significantly higher in the sitting (OR = 2.27, 95%CI = 0.63–8.18) and active groups (OR = 2.40, 95%CI = 0.66–9.75), though not the part-time mixed (OR = 0.86, 95%CI = 0.08–9.1). OPA pattern is a determinant of all-day PA during pregnancy. Future studies with larger samples should examine associations between pregnancy OPA patterns and pregnancy/infant health.

**Funding:** The MoM Health Study was funded to BBG by the American Heart Association (17GRNT3340016) with research registry, recruitment, and statistical support from the University of Pittsburgh Clinical and Translational Science Institute (NIH UL1TR0 0 0 0 05). The PRAMS Study was funded to KMW by the University of Iowa for Clinical and Translational Science (NIH UL1TR002537). The funders had no role in study design, data collection and analysis, decision to publish, or preparation of the manuscript. The funders had no role in study design, data collection and analysis, decision to publish, or preparation of the manuscript.

**Competing interests:** The authors have declared that no competing interests exist.

# Background

Physical activity is broadly recommended for people with uncomplicated pregnancies due to its many health benefits [1–3]. For example, the American College of Sports Medicine recommends that pregnant women engage in at least 150 minutes of moderate intensity physical activity, 75 minutes of vigorous intensity activity, or an equivalent combination each week to improve fitness and decrease excessive gestational weight gain (GWG), gestational diabetes (GDM), hypertensive disorders of pregnancy (HDP), and depressive symptoms [2]. However, these guidelines do not recommend that physical activity be accumulated in any particular domain; therefore, moderate-to-vigorous intensity physical activity (MVPA) that occurs during leisure, at work, doing household activities, or during transportation would all contribute equally to meeting weekly physical activity recommendations.

A better understanding of physical activity performed in the occupational domains during pregnancy is needed because the prevalence of women of childbearing age in the workforce has been increasing since the 1950s [4], and sixty-five percent of individuals continue to work during pregnancy [5]. Further, recent evidence suggests that occupational physical activity (OPA) may not be as beneficial as leisure time physical activity. A recent meta-analysis concluded that certain OPA exposures, such as prolonged standing, lifting, or heavy workloads, were associated with greater rates of preterm birth and other adverse pregnancy outcomes (APO) [6]. However, the authors cited several important limitations of the available studies, including measurement of OPA by retrospective self-report. Though research on OPA patterns in pregnancy is limited, more research is available on the effects of high OPA in non-pregnant adults. Several reviews have concluded that high levels of OPA have adverse or null associations with cardiovascular health outcomes and mortality [7, 8]. This phenomenon of opposing health impacts of leisure-time and OPA has been called the 'physical activity health paradox' [9]. However, a better understanding of whether such paradoxical associations are a concern during pregnancy is needed.

Another limitation of the available research in working pregnant populations is a focus on higher intensity OPA only. Other occupational exposures that could negatively impact pregnancy health include high levels of sitting or sedentary behavior (SB). In non-pregnant adults, high SB has been associated with increased cardiovascular disease risk, obesity, diabetes, and all-cause mortality [10, 11]. Breaking up prolonged SB has demonstrated acute benefits on cardiometabolic functions, including improved blood pressure and glucose disposal [12, 13], though there is a lack of research that has evaluated SB or interrupted SB during pregnancy. Yet, a small cohort study conducted by our group suggested that very high SB during pregnancy (~11 hours per day) was associated with worse maternal-fetal health outcomes [14–16]. Deskwork was a major determinant of high levels of SB in this pregnancy cohort [17], and desk-based jobs are similarly associated with higher levels of SB in the general population [18]. Clarification of how different OPA patterns, including both high OPA and high occupational SB, contribute to all-day activity and pregnancy health is needed to inform interventions and recommendations during pregnancy.

The purpose of this secondary analysis was to define OPA groups based on self-reported working status, OPA, and occupational SB across three trimesters of pregnancy and to determine if all-day device-measured activity levels differed across these groups. We further explored whether group status was associated with pregnancy and infant health outcomes. Outcomes investigated included resting blood pressure during each trimester, glucose from a glucose screen test, excessive GWG, HDP (including preeclampsia, gestational hypertension, and eclampsia), preterm birth, GDM, intrauterine growth restriction (IUGR), gestational age at delivery, and infant BMI z-scores. We hypothesized that device-measured physical activity,

including SB, light intensity physical activity (LPA), MVPA, and steps per day would differ across groups and that high levels of both OPA and occupational SB would be associated with less favorable pregnancy and infant outcomes.

## Methods

This secondary analysis included data from two cohort studies designed with similar methods: the Monitoring Movement & Health Study (MoM Health: Pittsburgh, PA) and the PRegnancy Activity Monitoring Study (PRAMS: Iowa City, IA). Study-specific research methods are described briefly below but have also been detailed previously [16].

### Participant characteristics

Including both studies, data were collected from 140 women (MoM Health: n = 120, PRAMS: n = 20) at three study visits that occurred between 8–13 weeks (first trimester), 22–24 weeks (second trimester), and 32–34 weeks (third trimester) gestation. Recruitment occurred for MoM Health from March 15, 2017 through September 11, 2018 and for PRAMS from July 31, 2018 through December 13, 2018. Study inclusion criteria included being 8–12 weeks pregnant at enrollment, 18–45 years old, and receiving care within the University of Pittsburgh Medical Center or a University of Iowa Health Care facility. Exclusion criteria included taking medications to treat pre-existing diabetes or hypertension, having a condition that severely limited physical activity, or participation in another research study involving a lifestyle intervention. For this current analysis, participants were excluded based on the criteria described in Fig 1. Of the original 140 participants, 131 were included in the current analysis (Pittsburgh: n = 111, Iowa: n = 20). Participants provided written informed consent prior to study enrollment with approval from the University of Pittsburgh and University of Iowa Institutional Review Boards for all research procedures.

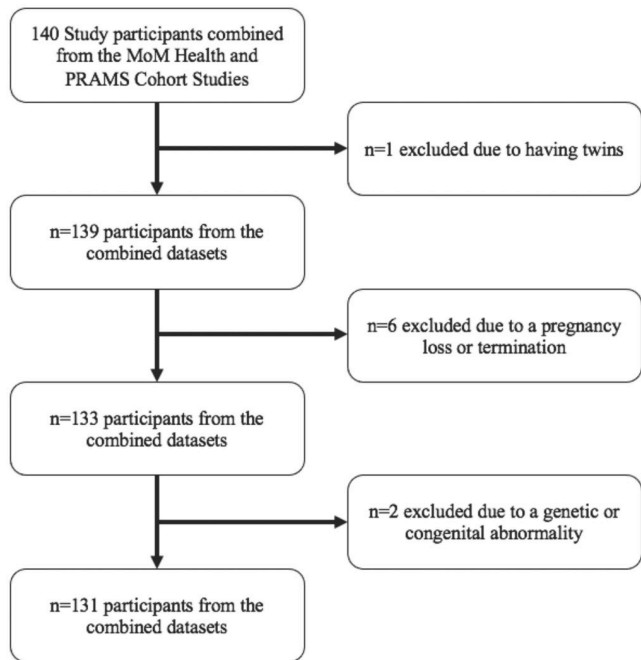

**Fig 1. Flowchart of exclusions from analytic sample.** This flowchart reports the number and reasons for excluding participants for this secondary analysis.

## Occupational physical activity

Participants completed the Pregnancy Physical Activity Questionnaire (PPAQ), a 32-item instrument assessing different types of physical activity, during each trimester. The PPAQ is well validated for OPA with high test-retest reliability (intraclass correlation coefficient = 0.93) and wide use in pregnant populations [19].

OPA level classifications were determined for participants who self-reported working for wages during their first study visit (n = 99). The five work-related PPAQ questions were then used to calculate duration of sitting, duration of standing or light walking, and duration of walking quickly [19]. The five OPA PPAQ questions queried the durations of the following activities while working: sitting, standing or slowly walking while carrying things, standing or slowly walking without carrying things, walking quickly while carrying things, and walking quickly without carrying things. Response options for each activity included 0, <0.5, 0.5–2, 2–4, 4–6, or ≥6 hours per day. The mid-point of each range was used to estimate the typical daily durations for each activity on working days (e.g., a response of 2–4 hours per day was scored as 3 hours per day). Standing or light walking was calculated by adding durations of standing or slowly walking while carrying things and standing or slowly walking without carrying things. Walking quickly was calculated by adding durations of walking quickly while carrying things and walking quickly without carrying things.

To calculate weekly durations of OPA in each category, daily proportions of work time spent sitting, standing and light walking, and walking quickly were calculated for each participant by dividing the time spent in each of these three OPA categories (reported 'per day') by the sum of all three OPA variables within each trimester. These proportions were multiplied by self-reported hours worked per week to provide weekly durations. This process was repeated for each trimester and then averaged across the trimesters. For participants that did not complete all three trimester study visits, averages of the available data were used for final analyses resulting in a final sample of n = 99 with OPA exposure during pregnancy. These three intensities of OPA were used to generate latent classes as described in the Statistical Analyses section of this manuscript.

## Leisure-time physical activity

Self-reported leisure-time physical activity was determined in metabolic equivalent hours per week (MET-h/week) based on responses to the PPAQ sports and exercise questions [19]. Test-retest reliability of the sports/exercise questions from this questionnaire has been found to be good with an intraclass correlation coefficient of 0.83 [19]. Volumes of leisure-time physical activity for each trimester were calculated using published algorithms [19, 20]. Trimester specific sample sizes for each group included the following: trimester 1 (sitting = 61, part-time mixed = 9, active = 28, non-working = 32), trimester 2: (sitting = 58, part-time mixed = 8, active = 27, non-working = 28), trimester 3: (sitting = 58 part-time mixed = 9, active = 26, non-working = 24).

## Device-based activity pattern assessment

All-day device-based physical activity was measured for one week during each trimester of pregnancy with average wear times for the activPAL and ActiGraph reported in S1 Table. Sample sizes for all-day activity variables by group and trimester are presented in S2 Table. Missing device-measured activity data was 7% in the first trimester, 12% in the second trimester, and 19% in the third trimester. For SB and steps, an activPAL3 micro was placed on the anterior, top third of the thigh with medical tape and worn 24 hours per day for seven days. Participants were asked to complete a diary to record if the device was removed and participant sleep onset

and offset times. Data were exported and processed with PAL Technologies software (version 7.2.38; PAL Technologies, Glasgow, Scotland) and participant diary entries were used to remove any reported non-wear periods and sleep time using standardized methods [21, 22]. In rare cases where participants did not report sleep times in the diary (4.7% of participants across the three visits), the device-based activity data was utilized to estimate sleep times. Sedentary time (% of waking wear time), prolonged sedentary time accumulated in bouts of greater than 30 minutes (% of waking wear time), and steps per day were quantified for each day of monitor wear. Estimates from valid days were averaged across the wear period and data were considered valid if there were at least four days of wear with a minimum of 10 wear hours each [23, 24].

MVPA was calculated as the percent of valid wear time and measured with an ActiGraph GT3X. The GT3X was worn around the waist with an elastic belt during waking hours for seven days. Adjustments in positioning of the belt to below the abdomen were made with pregnancy progression, but the monitor always remained aligned vertically with the right knee [25]. MVPA was quantified using 1-minute epochs and ActiLife software (version 6.12.2; ActiGraph, LLC, Pensacola, FL), where the Choi algorithm defined valid wear time and epochs that were ≥2690 counts per minute were summed to determine daily MVPA [23, 26]. Similar to the activPAL, data was considered valid with a minimum of four days of wear for at least 10 hours per day [24].

Average daily time spent in LPA was estimated using daily averages from the activPAL and GT3X as described above. LPA was calculated as follows: 100% of waking wear time minus [percent of time spent sedentary from activPAL + percent of time spent in MVPA from GT3X].

## Health outcomes

Resting blood pressure was calculated as the average of duplicate measures after a 10-minute rest period (with a third measurement taken if SBP/DBP differed by greater than or equal to 10/6 mmHg) in MoM Health or a 5-minute rest with the average of the second and third of three measurements in PRAMS. Both studies measured blood pressure in each trimester of pregnancy using an OMRON HEM-705 oscillometric device.

All other pregnancy and infant outcomes were abstracted from medical records. Following birth, medical records were reviewed independently by two research team members, with a maternal-fetal medicine physician consultation if necessary. Pregnancy outcomes included glucose from a 50g glucose screening test, gestational age at delivery, excessive GWG, and APOs (HDP [including gestational hypertension, preeclampsia, and eclampsia], IUGR, GDM, and preterm birth). GWG was calculated as the difference between weight at delivery and pre-pregnancy weight with excessive GWG classified using the Institute of Medicine guidelines [27]. During medical record reviews, it was determined that a participant experienced an APO with a physician diagnosis based on the American College of Obstetrics and Gynecology definitions [28, 29]. A composite, binary APO outcome was generated and included participants that experienced any APO (n = 30). Two participants were excluded from APO analyses due to missing medical record data. Infant outcomes obtained from the birth medical record included birthweight, height, and sex, which were utilized to calculate infant BMI z-score values [30].

## Statistical analyses

All statistical analyses were performed using Stata BE version 17 [31]. As mentioned previously, the three-intensity specific OPA duration variables were used in a latent class analysis to

categorize participants who worked into groups. Three groups was the best fitting model based on Bayesian information criterion value, Akaike information criterion value, group sample sizes, and clinical meaningfulness/model interpretability (S3 Table). The resultant three OPA latent classes included a sitting, part-time mixed activity, and active group. Average hours per week spent in the three OPA intensities differed by latent class groups (p<0.001) with the part-time mixed group working less hours (14.6 hours per week) compared to the sitting (40.4 hours per week) and active groups (40.0 hours per week) as presented in Fig 2. Examples of occupations for the group that spent the majority of their workday sitting (i.e., the sitting group; n = 61) included administrative positions, university faculty, research staff, finance employees, and information technology positions. The group that spent their workday in a mixture of OPA including sitting, standing and light walking, and walking quickly (i.e., the part-time mixed group; n = 9) included a pre-school teacher, interpreter, and some healthcare workers (e.g., nurse practitioner and social worker). The third group that spent their workday with less sitting and more activity (i.e., the active group; n = 29) included caregivers, service workers (e.g., waitresses), and hospital staff (e.g., patient care technicians, aides, and nurses). A fourth group was defined independently from the latent class analysis and included participants that self-reported not working for wages during their first study visit (n = 32) and, as such, had no OPA.

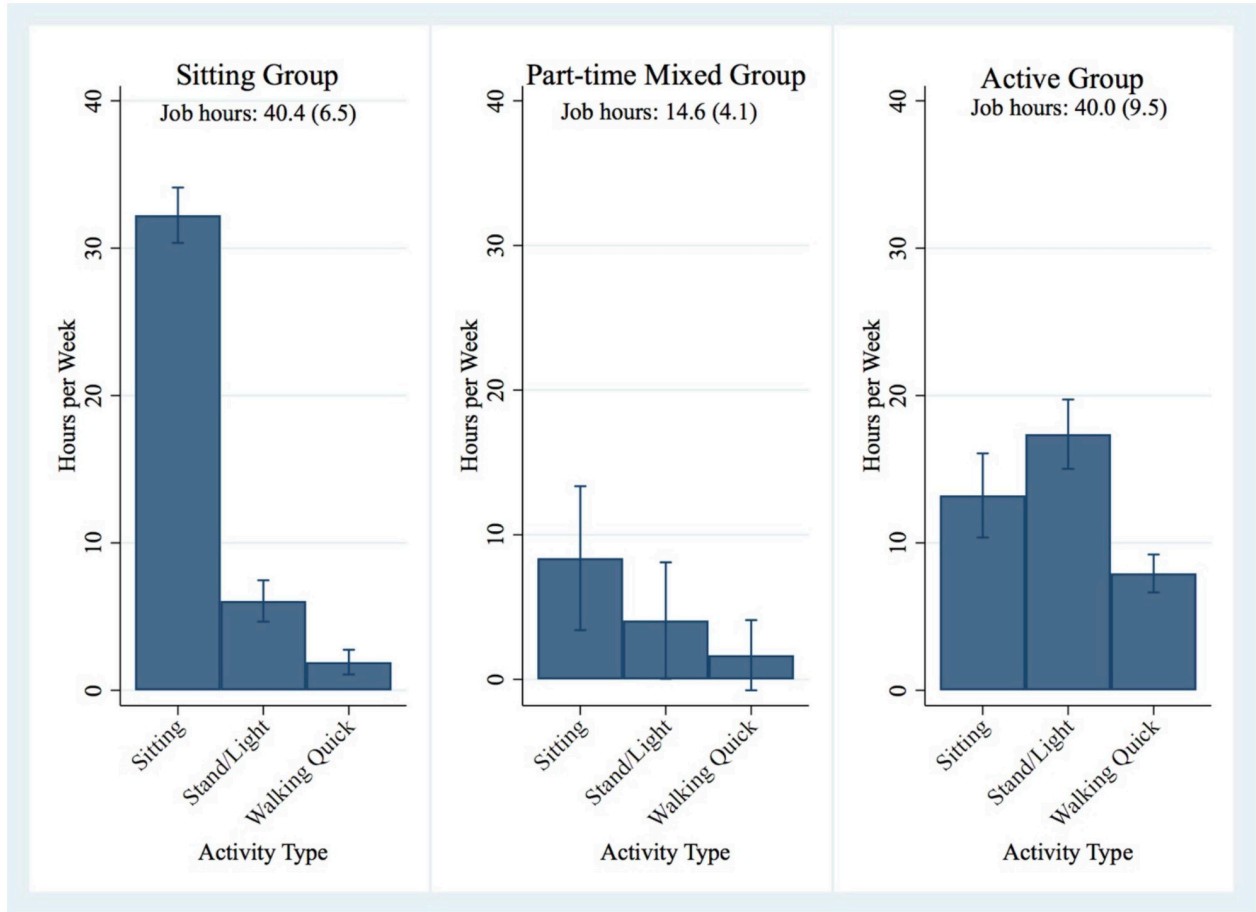

**Fig 2. Activity compositions across groups among working participants.** The bar graphs display the average number of hours per week working participants spent in different OPA intensities based on latent class identified groups. Job hours worked per group are listed below the group titles as mean (SD).

One-way ANOVA or chi-squared analyses were used to compare participant characteristics between the four groups. Linear regression models with likelihood ratio tests evaluated differences in device-based activity variables across groups. Mixed effects models evaluated whether resting blood pressure (measured in each trimester) differed across groups, and the overall variability explained by group was evaluated using likelihood ratio tests. Linear or logistic regression models with likelihood ratio tests evaluated associations of the four groups with pregnancy and infant outcomes. For regression models, the non-working group was chosen as the reference group to evaluate risk associated with group membership. When significant differences were detected, pairwise post hoc analyses with Bonferroni adjustment for multiple comparisons were conducted to determine the presence of statistical difference between group differences in all-day activity. Logistic regression models and likelihood ratio tests were not conducted for outcomes where there were less than 10 events. All regression models were adjusted for age, education, race, and pre-pregnancy BMI. Job hours were considered as a covariate but were ultimately not included due to collinearity with the group variables that resulted in variance inflation factors greater than 10. Study site was also considered as a covariate but was ultimately not included due to minimal differences in participant characteristics between sites, concerns of parsimony due to the small sample, and minimal differences in sensitivity analyses that included site.

## Results

### Participants

There were 131 participants from the combined datasets used in final analyses, including 99 that self-reported working during their pregnancy and 32 that did not. S4 Table displays participant characteristics overall and by site. The only difference between sites was that self-reported leisure-time PA was higher in the MoM health cohort compared to PRAMS during the second (p<0.001) and third trimester visits (p = 0.008). Table 1 reports participants characteristics by latent class defined groups. Age differed significantly between groups (p = 0.003)

**Table 1. Participant characteristics by group.**

|  | Total (n = 131) | Sitting (n = 61) | Part-time Mixed (n = 9) | Active (n = 29) | Non-working (n = 32) | p-value |
|---|---|---|---|---|---|---|
| **Age** (years) | 30.9 ± 4.9 | 32.2 ± 4.1 | 31.4 ± 4.0 | 30.6 ± 5.4 | 28.3 ± 5.1 | **0.003** |
| **Pre-Pregnancy BMI** (kg/m$^2$) | 26.8 ± 6.7 | 26.3 ± 5.9 | 24.1 ± 3.9 | 29.1 ± 8.0 | 26.5 ± 7.3 | 0.158 |
| **Race** |  |  |  |  |  | 0.166 |
| White | 101 (77.1%) | 53 (86.9%) | 7 (77.8%) | 21 (72.4%) | 20 (62.5%) |  |
| Black | 23 (17.6%) | 5 (8.2%) | 2 (22.2%) | 6 (20.7%) | 10 (31.2%) |  |
| Other | 7 (5.3%) | 3 (4.9%) | 0 (0.0%) | 2 (6.9%) | 2 (6.2%) |  |
| **Education** |  |  |  |  |  | **<0.001** |
| Less than a Bachelor's Degree | 44 (33.6%) | 9 (14.8%) | 2 (22.2%) | 13 (44.8%) | 20 (62.5%) |  |
| Bachelor's Degree or Higher | 87 (66.4%) | 52 (85.2%) | 7 (77.8%) | 16 (55.2%) | 12 (37.5%) |  |
| **Self-Reported Leisure-time PA (MET-h/week)** |  |  |  |  |  |  |
| **1$^{st}$ Trimester** (n = 130) | 17.0 (14.9) | 18.0 (15.2) | 22.1 (18.3) | 15.0 (12.0) | 15.6 (14.9) | 0.535 |
| **2$^{nd}$ Trimester** (n = 121) | 15.3 (13.9) | 15.1 (14.4) | 21.2 (18.0) | 14.0 (11.9) | 15.3 (13.5) | 0.647 |
| **3$^{rd}$ Trimester** (n = 117) | 11.9 (13.4) | 12.5 (14.2) | 13.6 (13.7) | 8.4 (10.0) | 13.4 (14.9) | 0.521 |

Data presented as n (%) or mean (SD); Abbreviations: BMI = body mass index, SD = standard deviation, PA = physical activity, MET = metabolic equivalents, OPA = occupational physical activity; p-values reflect a hypothesis test across groups using one-way ANOVA for continuous variables or chi-square for categorical variables; Bold p-values indicate statistical significance

such that non-working individuals were younger (mean = 28.3 years old, sd = 5.1) and those in the sitting group were older (mean = 32.2 years old, sd = 4.1) as compared to the part-time mixed (mean = 31.4 years old, sd = 4.0) and active (mean = 30.6 years old, sd = 5.4) groups. Education levels differed between groups (p = <0.001) with individuals in the active and non-working groups being more likely to have less than a bachelor's degree (active = 44.8%; non-working = 62.5%) compared to those with high sitting occupations or a mixture of work activities (sitting = 14.8%; part-time mixed = 22.2%)." Pre-pregnancy BMI, race, and self-reported leisure-time physical activity did not differ between groups.

## Activity pattern results

Table 2 presents device-based activity measures by group across trimesters of pregnancy. SB was significantly different across groups in each trimester (all p<0.05) with the active group having lower SB compared to the sitting group (*post hoc* p<0.05). LPA also differed across groups in each trimester (all p<0.03), with the active group having higher LPA compared to the sitting group (*post hoc* p<0.05). Steps per day differed across group in each trimester (all p<0.004), with the active group having higher steps per day compared to the other three groups (*post hoc* p<0.05). MVPA differed by group only in the second trimester (p = 0.034),

**Table 2. Device-based all-day activity quantification by group.**

| | Sitting (n = 61) | Part-time Mixed (n = 9) | Active (n = 29) | Non-Working (n = 32) | p-value** |
|---|---|---|---|---|---|
| **SB (% of activPAL wear time)** | | | | | |
| 1st Trimester[a] | 67.9 (8.3) | 63.1 (12.1) | 59.1 (10.0) | 65.5 (11.6) | **<0.001** |
| 2nd Trimester[a] | 65.4 (8.2) | 67.0 (10.1) | 58.1 (8.9) | 62.4 (11.5) | **0.004** |
| 3rd Trimester[a] | 65.3 (9.1) | 65.7 (8.9) | 58.7 (10.0) | 62.6 (11.2) | **0.007** |
| **Prolonged SB (≥30-minute bouts) (% of acticPAL wear time)** | | | | | |
| 1st Trimester | 37.0 (11.6) | 31.6 (15.5) | 30.9 (9.6) | 33.6 (13.9) | 0.077 |
| 2nd Trimester | 35.9 (11.0) | 34.1 (12.0) | 30.1 (11.4) | 32.0 (12.0) | 0.184 |
| 3rd Trimester | 35.9 (10.9) | 34.3 (11.2) | 30.5 (10.6) | 31.9 (14.1) | 0.071 |
| **LPA (% of activPAL and ActiGraph wear time)** | | | | | |
| 1st Trimester[a] | 28.3 (7.8) | 32.0 (12.3) | 36.8 (9.5) | 31.8 (11.1) | **<0.001** |
| 2nd Trimester[a] | 30.8 (8.0) | 30.0 (10.7) | 37.7 (8.8) | 35.1 (11.5) | **0.008** |
| 3rd Trimester[a] | 31.8 (8.8) | 32.1 (10.5) | 38.1 (9.6) | 35.0 (11.5) | **0.021** |
| **MVPA (% of ActiGraph wear time)** | | | | | |
| 1st Trimester | 3.8 (1.9) | 3.1 (1.2) | 4.1 (1.7) | 2.9 (1.9) | 0.135 |
| 2nd Trimester | 3.8 (1.9) | 3.8 (1.7) | 4.4 (1.8) | 3.3 (1.6) | **0.034** |
| 3rd Trimester | 3.0 (1.9) | 2.8 (1.4) | 3.3 (1.6) | 2.9 (1.7) | 0.602 |
| **Steps per day** | | | | | |
| 1st Trimester[b] | 7,274 (2,591) | 7,594 (2,239) | 8,891 (2,582) | 6,587 (2,456) | **<0.001** |
| 2nd Trimester[b,c] | 7,483 (2,697) | 6,380 (2,075) | 8,899 (2,811) | 7,006 (2,312) | **0.003** |
| 3rd Trimester[a,c] | 6,608 (2,298) | 6,439 (1,276) | 8,167 (2,160) | 6,867 (2,816) | **0.002** |

Data are presented as mean (SD); n values represent maximum sample size per group;

** Likelihood ratio test; Adjusted for age, pre-pregnancy BMI, race, and education level;

Superscript letters correspond to pairwise differences observed in *post hoc* testing with Bonferroni adjustment for multiple comparisons:

[a] = significant difference between sitting versus active group,

[b] = significant difference between non-working versus active group,

[c] = significant difference between mixed versus active group

Abbreviations: MVPA = moderate-to-vigorous physical activity, LPA = light physical activity, SB = sedentary behavior, OPA = occupational physical activity; Bold p-values indicate statistical significance

though pairwise differences were not statistically different after adjustment for multiple comparisons. No differences were observed across groups for MVPA during the first and third trimester or prolonged SB in any trimester (all p>0.07).

## Pregnancy health and outcome results

No differences were observed across groups for maternal clinical outcomes (glucose screen and blood pressure) or infant outcomes (gestational age, BMI z-score) (Table 3). Odds of any APO (n = 30), HDP (n = 23), and excessive GWG (n = 60) comparing each group to the non-working group are displayed in Fig 3. Preterm birth, IUGR, and GDM were less common (<10 events) and not observed in each group. Though these were included in the composite score of any APO, they were not individually analyzed (frequencies by group are reported in S5 Table). Despite the nonsignificant findings, the sitting and active groups consistently had higher values for the glucose screen, blood pressure, and greater odds of any APO, HDP, and excessive GWG as compared to the non-working group, while the part-time mixed group appeared to have non-significantly lower values and risk of some pregnancy outcomes compared to the non-working group.

## Discussion

Pregnant participants that worked during our study had three typical OPA patterns: predominantly sitting, mixed activities with lower (part-time) work hours, or a more active occupation. Our hypothesis that OPA group membership is an important determinant of all-day activity patterns in working individuals was supported. Those who self-reported predominantly sitting at work spent a higher percentage of the day sitting and typically had fewer steps compared to the part-time mixed, active, and non-working groups. The part-time mixed group was characterized by less overall working hours and intermediate amounts of sitting, LPA, and steps. The active group had significantly higher amounts of LPA and steps per day during all three trimesters and higher MVPA during the second trimester. In our exploratory analyses, maternal resting blood pressure, glucose, and infant outcomes did not differ amongst the groups. However, although not significantly different and limited by our small sample size, our data provides preliminary support of the hypothesis that high OPA and high occupational SB could be associated with less favorable outcomes. These associations should be investigated further.

Our findings suggesting that self-reported OPA was an important determinant for all-day activity patterns in a pregnant population align with findings in similar studies within large cohorts of non-pregnant adults. In the Coronary Artery Risk Development in Young Adults

**Table 3. Maternal clinical outcomes and infant outcomes by groups across pregnancy trimesters.**

|  | N | Sitting | Part-time Mixed | Active | Non-Working | p-value** |
|---|---|---|---|---|---|---|
|  |  | **Maternal Clinical Outcomes** | | | | |
| **Glucose Screen** (mg/dL) | 118 | 113.2 (105.3, 121.0) | 105.7 (83.7, 127.7) | 113.5 (102.3, 124.7) | 104.2 (93.4, 115.0) | 0.749 |
| **SBP** (mmHg) | 131 | 113.2 (111.1, 115.2) | 110.1 (104.8, 115.4) | 115.6 (112.6, 118.4) | 111.5 (108.6, 114.4) | 0.619 |
| **DBP** (mmHg) | 131 | 67.8 (66.4, 69.2) | 65.9 (62.3, 69.5) | 68.3 (66.2, 70.3) | 65.6 (63.8, 67.8) | 0.906 |
|  |  | **Infant Outcomes** | | | | |
| **Gestational Age** (weeks) | 126 | 38.6 (38.4, 39.3) | 40.2 (39.1, 41.3) | 39.1 (38.5, 39.7) | 38.8 (38.2, 39.4) | 0.093 |
| **Infant BMI z-score** | 121 | -0.49 (-0.82, -0.15) | -0.76 (-1.6, 0.69) | -0.18 (-0.65, 0.29) | -0.47 (-0.94, -0.52) | 0.895 |

Data are presented as mean (95% CI); Abbreviations: SBP = systolic blood pressure, DBP = diastolic blood pressure, BMI = body mass index, OPA = occupational physical activity, 95% CI = 95% confidence interval;

** Likelihood ratio test; Adjusted for age, pre-pregnancy BMI, race, and education level

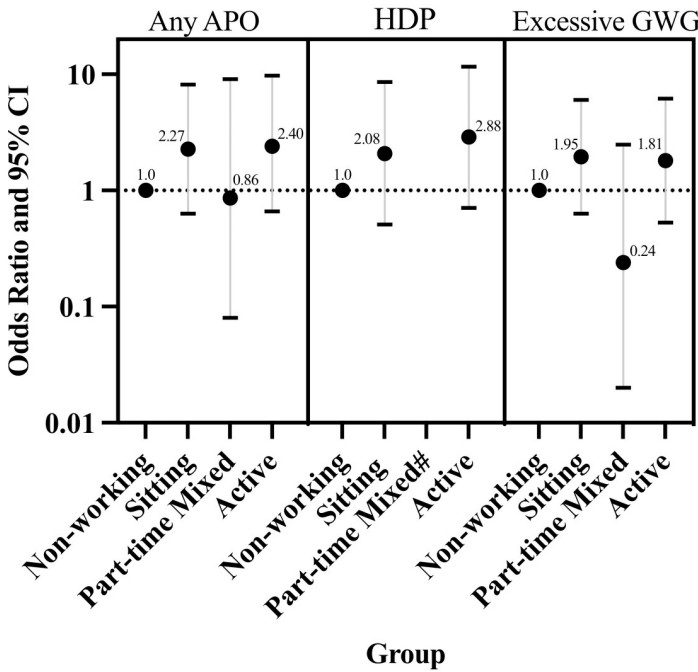

**Fig 3. Adjusted odds of any APO HDP and excessive GWG by group.** Abbreviations: APO = adverse pregnancy outcome (n = 129), HDP = hypertensive disorders of pregnancy (n = 127), GWG = gestational weight gain (n = 124); # there were no HDP events in the part-time mixed group, so odds ratio was not calculated; Data adjusted for age, pre-pregnancy BMI, race, and education level.

(CARDIA) cohort study, our group found that LPA and SB measured by a waist accelerometer differed between employment status and occupational groups (all p<0.001) [18]. Similarly in the National Health and Nutrition Examination Survey (NHANES), Steeves et al. concluded that device-measured all-day physical activity levels in non-pregnant adults differed between low, intermediate, and high OPA groups [32]. Our data expands this research by supporting the substantial role of occupation in all-day activity patterns during pregnancy. This is as an important finding as many pregnant people remain in the workforce throughout pregnancy, including, at least 56% of the United States pregnant population as documented by census data [5] and 75% of the participants in our sample. This high percentage of working pregnant individuals highlights the need for considering OPA patterns in the assessment of guidelines for physical activity during pregnancy.

Yet, the impact of high OPA during pregnancy is poorly understood. Limited research investigates the association of occupational status and OPA patterns on pregnancy health and the few studies available typically have poor measurement of OPA (i.e., by self-report only, or only measuring a single type of activity such as lifting) [6]. Despite noting these limitations, a recent meta-analysis by Cai et al. found that a heavy physical workload appeared to be associated with higher odds of preterm birth (OR = 1.23, 95% CI = 1.07–1.41), miscarriage (OR = 1.49, 95% CI = 0.91–2.45), preeclampsia (OR = 1.30, 95% CI = 0.69–2.43), and small for gestational age (OR: 1.34, 95% CI: 1.03–1.73) [6]. Another study published after the Cai et al. meta-analysis found that self-reported occupational standing or walking for greater than 30 hours per week significantly increased the risk of preterm birth indicated due to gestational hypertension (OR = 2.09, 95% CI = 1.00–4.97) [33]. While results may appear inconsistent with the null findings of our study; that are limited in power due to our small sample and

event rates, our finding of a non-significant but more than two-fold higher risk of any APO and HDP in the active vs. non-working group could be coherent with the previous findings.

Despite the limited data in pregnancy, prior research has investigated the potential negative impact of high OPA on cardiovascular and other health outcomes in non-pregnant adults. The 'physical activity health paradox' hypothesis suggests that leisure time physical activity and OPA have different impacts on health. Leisure time physical activity is typically higher in intensity, acute in nature, and is followed by an adequate recovery period that results in the well documented benefits of physical activity on cardiovascular health. Conversely, OPA is typically prolonged and at a lower intensity [9], leading to sustained elevated heart rate and blood pressure during and possibly following OPA. High OPA has also been associated with autonomic dysfunction and increased circulating inflammatory markers [34–36]. Altogether, OPA may have several negative cardiovascular impacts that promote adverse cardiovascular processes [37]. As pregnancy is a cardiovascular challenge in the absence of OPA, the above-mentioned physiological effects of OPA may put further stress on the cardiovascular system during pregnancy and could lead to the detrimental pregnancy and infant health outcomes observed in the previous meta-analysis and in non-significant associations in our study [6]. Greater understanding of these mechanisms and associations is an area in need of further research.

Research on associations between high SB and pregnancy health outcomes is even more scarce, though the relationship between high SB and poor health outcomes is increasingly recognized in non-pregnant adults. High levels of SB, especially when accumulated in prolonged bouts (i.e. >30 min), is consistently associated with a higher risk of long-term health outcomes such as cardiovascular disease [38] and all-cause mortality [39]. Suggested mechanisms from previous studies for the negative cardiovascular impacts from prolonged sitting include reduced venous return and vascular dysfunction in the lower limbs, resulting in blood pooling that stresses the cardiovascular system [40, 41]. These physiological effects would also be expected to occur and could be exacerbated in pregnancy. In our small MoM Health cohort (n = 120) that made up the majority of the sample for the current secondary analysis, we found that being in the highest device-measured sedentary behavior group (~11 hours per day) was associated with higher blood pressure [14] and higher odds of adverse pregnancy outcomes during pregnancy [16]. Another study by Meander et al. examined self-reported SB with pregnancy health and concluded that there was an increased odds of participants experiencing a HDP when sitting more than 7 hours per day (OR = 1.65, 95% CI = 1.03–2.62) compared to those who spent less than 7 hours per day sitting during pregnancy [42]. [NO_PRINTED_-FORM]Taken together, specific research on the associations of overall and occupational SB with pregnancy and infant health outcomes is an area in need of more research.

Strengths of this secondary analysis include prospective OPA measurements using a validated questionnaire, the device-based measures of all-day activity, and medical record abstractions to measure pregnancy and infant health outcomes. Although the device-based monitors used for this study are best practice measures to determine SB and MVPA, [43, 44] LPA was calculated based on the two devices rather than directly measured, which may have introduced additional error for the measurement of LPA. Other limitations of this study included the small sample size, self-report measures of OPA, and a homogenous sample of mostly white and highly educated pregnant people. A latent class analysis determined three distinct groups for this study based on the work-related questions in the PPAQ, but future studies should investigate more specific types of OPA such as heavy lifting and bending. Lastly, the latent class analysis revealed a group with mixed activity and part-time working hours. This group differed from the other groups in both average working hours (the other groups were typically full time) and activity levels. Though we hoped to statistically adjust for working hours in

regression models, high collinearity resulted in variance inflation between groups and working hours and precluded us from including job hours as a covariate. Thus, the contribution of the OPA exposure and work duration could not be disentangled, and results should be interpreted with that caveat in mind. Future studies should include larger, more diverse samples and measure OPA using activity monitors and more specific methods. It would also be beneficial to investigate if there is any relationship between prolonged or interrupted occupational SB and OPA on pregnancy and infant outcomes.

## Conclusions

Our findings advance our understanding of the importance of OPA as a determinant of all-day activity in pregnancy. Combined with population trends where a high proportion of pregnant individuals are exposed to OPA and occupational SB, a better understanding of the impact of certain activities performed throughout the workday on pregnancy and infant health is necessary. Such data could inform health promotion and risk mitigation for working pregnant individuals. Though not statistically significant, the public health significance of the possibly elevated risk of some adverse pregnancy outcomes in the high OPA and high SB occupational groups calls for more rigorous research in this area. Given the well-established benefits of leisure time physical activity, further investigation of the comparatively less clear impact of OPA patterns on pregnancy health outcomes is needed to support optimal health for those that are continuing to work during pregnancy.

## Supporting information

**S1 Table. Average wear times for the activPAL and ActiGraph monitors by trimester.** Data are presented as mean (sd).
(DOCX)

**S2 Table. Sample sizes for all-day activity data by group as presented in Table 2.** Data are presented as n values; Abbreviations: MVPA = moderate-to-vigorous physical activity, LPA = light physical activity, SB = sedentary behavior.
(DOCX)

**S3 Table. Statistics for different latent class models from one to four classes.** Abbreviations: BIC = Bayesian Information Criteria, AIC = Akaike Information Criterion.
(DOCX)

**S4 Table. Participant characteristics by site.** Data presented as n (%) or mean (SD); Abbreviations: BMI = body mass index, SD = standard deviation, PA = physical activity, MET = metabolic equivalents, p-values reflect a hypothesis test across study sites using one-way ANVOA for continuous variables or chi-square for categorical variables; Bold p-values indicate statistical significance.
(DOCX)

**S5 Table. Frequency and odds of experiencing an APO by group.** Abbreviations: APO = any adverse pregnancy outcome, HDP = hypertensive disorders of pregnancy, GWG = gestational weight gain, IUGR = intrauterine growth restriction, GDM = gestational diabetes mellitus 95% CI = 95% confidence interval; **Likelihood ratio test evaluating all groups; Adjusted for age, pre-pregnancy BMI, race, and education level; ***Likelihood ratio test excluding the part-time mixed group due to no events;—indicates odds ratios and likelihood test were not conducted due to low number of events.
(DOCX)

**S1 Dataset. This is the minimal anonymized data set that was used for analyses in this manuscript.** The codebook is included as a second ab within the excel document.
(XLSX)

## Author Contributions

**Conceptualization:** Tyler Quinn, Kara M. Whitaker, Bethany Barone Gibbs.

**Data curation:** Alexis Thrower, Tyler Quinn, Melissa Jones, Bethany Barone Gibbs.

**Formal analysis:** Alexis Thrower, Bethany Barone Gibbs.

**Funding acquisition:** Kara M. Whitaker, Bethany Barone Gibbs.

**Investigation:** Kara M. Whitaker, Bethany Barone Gibbs.

**Methodology:** Alexis Thrower, Melissa Jones, Kara M. Whitaker, Bethany Barone Gibbs.

**Project administration:** Melissa Jones, Kara M. Whitaker, Bethany Barone Gibbs.

**Resources:** Melissa Jones.

**Supervision:** Bethany Barone Gibbs.

**Writing – original draft:** Alexis Thrower.

**Writing – review & editing:** Tyler Quinn, Melissa Jones, Kara M. Whitaker, Bethany Barone Gibbs.

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
