## [Decision Letter · Decision Letter 0]

20 Oct 2023

PONE-D-23-20019Occupational physical activity as a determinant of daytime activity patterns and pregnancy and infant healthPLOS ONE

Dear Dr. Thrower,

Thank you for submitting your manuscript to PLOS ONE. After careful consideration, we feel that it has merit but does not fully meet PLOS ONE’s publication criteria as it currently stands. Therefore, we invite you to submit a revised version of the manuscript that addresses the points raised during the review process.

We look forward to receiving your revised manuscript.

Kind regards,

Zhaoxia Liang

Academic Editor

PLOS ONE

Journal Requirements:

Additional Editor Comments :

The study bases on two prospective cohort studies to explore the association between occupational physical activity and SB, and pregnancy/infant health outcomes. The current study is interesting and add new evidence related to this topic. But I have a few concerns:

1. Participant Characteristics: As the two study facilities are different in the two institutions, Table 1 should be reported for each institution. And it would be easier to understand if you could provide the overall average and other information in writing.

2. Suggest to add a flow chart of inclusion and exclusion criteria, which will be easier to understand.

3. Result: The p value in the same table seems to be analyzed in different way(such as line 268), Please clarify blow the table.

Reviewers' comments:

Reviewer's Responses to Questions

**Comments to the Author**

1. Is the manuscript technically sound, and do the data support the conclusions?

Reviewer #1: Partly

2. Has the statistical analysis been performed appropriately and rigorously? 

Reviewer #1: Yes

3. Have the authors made all data underlying the findings in their manuscript fully available?

Reviewer #1: No

4. Is the manuscript presented in an intelligible fashion and written in standard English?

Reviewer #1: Yes

5. Review Comments to the Author

Reviewer #1: This manuscript examines occupational physical activity as a determinant of total physical activity and pregnancy and infant health outcomes. This topic is interesting and the manuscript is generally well written. The following is a list of my comments for the authors to clarify/consider.

Abstract:

Line 29, "and hours and OPA", is there a typo?

Line 40, suggest rephrasing "objective monitors"

Line 41, It is unclear what the p value in this sentence is for. There seems to be multiple comparisons.

Methods:

line 124 and Line 174, intraclass correlation coefficient was provided to indicate reproducibility. Isn't test-rest reliability a better term for reproducibility? Clarify if this was used in the original publication.

Line 139, clarify how proportions of work time spent in each category were calculated. The answer options in the questionnaire were in hours per day. What was the reason to convert to proportions?

Line 143, when participants did not complete all visits, why were the averages of available data used, rather than the last available data, especially if the last visit was missing. Were missing data distributed evenly across the visits?

Paragraph starting 146, have the authors considered moving this to the "Statistics" section? It talks about analysis.

Line 156, "in a mixture all three activities", clarify this phrase

Line 175 and other places: what about putting the sample sizes in a table or figure? The way they are presented in separate paragraphs is confusing and hard to grasp.

Line 186, it seems sleep time was determined based on diaries. For those who did not report sleep times and days they did not report, how was it determined? What was the percentage of women reporting sleep in the diary on a daily basis?

Line 236 and other places, suggest be careful when using "objective activity". Light intensity PA was calculated using data from 2 monitors.

Results:

Please report wear time of both monitors, number of days and number of hours worn per day.

Were days only one monitor was worn included in calculations? Was light intensity PA calculated using the same day of the 2 monitors or the averages of the few days each monitor was worn?

Table 2, clarify "% of wear time" is % of which monitor's wear time.

Line 268 and other places, are the p values in the table for the overall ANOVA test? Were follow-up tests done for pairwise comparison? The sentence seems to about comparing MVPA in the active group compared to other groups, which is different from the overall ANOVA test. Similar interpretations were made in several other places.

Were leisure time PA reported in Results, since it was talked about in Methods.

Why was study site not adjusted?

Discussion

It mentions that the "mixed actvities" group had "part-time" work hours. It is not clear this was actually included in Results. Please clarify. Perhaps provide number of hours worked for each group? Especially this was not adjusted in the models.

Line 376, was "leisure-time SB" correct in this sentence?

Need to discuss limitations of using 2 monitors. Would any category of time spent be under/overestimated?

6. PLOS authors have the option to publish the peer review history of their article (what does this mean?). If published, this will include your full peer review and any attached files.

Reviewer #1: No

---

## [Author Response · Author response to Decision Letter 0]

21 Nov 2023

Dear Dr. Liang,

Thank you for the opportunity to submit a revised draft of our manuscript titled, ‘Occupational physical activity as a determinant of daytime activity patterns and pregnancy and infant health’ to PLOS ONE. We appreciate you and the reviewer taking the time and effort to provide valuable feedback on this manuscript. For each comment, we have responded in a point-by-point fashion and incorporated changes to reflect the suggestions. All changes are detailed following the editor’s and reviewer’s comments below and tracked within the manuscript Word document. We feel these edits have improved the manuscript, and we hope you will too. Clarification was requested for the funding information related to this manuscript. The funding statement is below:

“The MoM Health Study was funded to BBG by the American Heart Association (17GRNT3340016) with research registry, recruitment, and statistical support from the University of Pittsburgh Clinical and Translational Science Institute (NIH UL1TR0 0 0 0 05). The PRAMS Study was funded to KW by the University of Iowa. The funders had no role in study design, data collection and analysis, decision to publish, or preparation of the manuscript. The funders had no role in study design, data collection and analysis, decision to publish, or preparation of the manuscript.”

In addition to the comments below, additional edits were made to the formatting of the manuscript to better reflect the PLOS ONE guidelines and the reference list was checked to ensure it is complete and correct. Changes made to the reference list included updated URLs for references 2 and 5. Finally, all figures were uploaded to the Preflight Analysis and Conversion Engine (PACE) digital diagnostic and reuploaded to this submission. We look forward to hearing from PLOS ONE regarding this manuscript resubmission and to continue the peer review process. 

Sincerely,

Alexis Thrower, MS

Graduate Student Researcher 

Doctoral Student in Exercise Physiology

Department of Exercise Physiology 

Comments from the editor

Comment 1: Participant Characteristics: As the two study facilities are different in the two institutions, Table 1 should be reported for each institution. And it would be easier to understand if you could provide the overall average and other information in writing.

Response: We agree that reporting participant characteristics by site would improve this manuscript. We added a Supplemental Table 4 that includes site-specific reporting and comparison of characteristics. Since the majority of characteristics were not significantly different across sites, the original Table 1 already included 7 columns to stratify by OPA exposure status, and we conducted sensitivity analyses for site we decided to make this information supplemental instead of adding an additional table to the manuscript. Below in bold, blue font, you will find the text added to the first paragraph of the results section and the Supplemental Table 4. Additionally, we provided averages in-text based on Table 1 in the first paragraph of the Results section and below.

Results (Lines 423-424, 426-442):

“Supplemental Table 4 displays participant characteristics overall and by site. The only difference between sites was that self-reported leisure-time PA was higher in the MoM health cohort compared to PRAMS during the second (p<0.001) and third trimester visits (p=0.008).”

 Overall

(n=131) Pittsburgh Site 

(MoM Health: n=111) Iowa Site 

(PRAMS: n=20) p-value

Age (years) 30.9 ± 4.9 30.9 (5.0) 30.5 (4.3) 0.687

Pre-Pregnancy BMI (kg/m2) 26.8 ± 6.7 26.4 (6.8) 29.2 (5.6) 0.086

Race 0.077

White 101 (77.1%) 82 (73.9) 19 (95.0) 

Black 23 (17.6%) 23 (20.7) 0 (0) 

Other 7 (5.3%) 6 (5.4) 1 (5.0) 

Education 0.510

Less than a Bachelor’s Degree 44 

(33.6%) 36 (32.4) 8 (40.0) 

Bachelor’s Degree or Higher 87 

(66.4%) 75 (67.7) 12 (60.0 

Self-Reported Leisure-time PA (MET-h/week)

1st Trimester (n=130) 17.0 

(14.9) 17.7 (15.3) 13.0 (9.1) 0.198

2nd Trimester (n=121) 15.3 

(13.9) 17.2 (14.2) 5.0 (4.3) <0.001

3rd Trimester (n=117) 11.9 

(13.4) 13.3 (14.1) 4.5 (4.6) 0.008

“Table 1 reports participants characteristics by latent class defined groups. Age differed significantly between groups (p=0.003) such that non-working individuals were younger (mean= 28.3 years old, sd=5.1) and those in the sitting group were older (mean=32.2 years old, sd=4.1) as compared to the part-time mixed (mean=31.4 years old, sd=4.0) and active (mean=30.6 years old, sd=5.4) groups. Education levels differed between groups (p=<0.001) with individuals in the active and non-working groups being more likely to have less than a bachelor’s degree (active= 44.8%; non-working=62.5%) compared to those with high sitting occupations or a mixture of work activities (sitting=14.8%; part-time mixed=22.2%).”

Supplemental Table 4. Participant Characteristics by Site

Data presented as n (%) or mean (SD); Abbreviations: BMI=body mass index, SD= standard deviation, PA=physical activity, MET=metabolic equivalents, p-values reflect a hypothesis test across study sites using one-way ANVOA for continuous variables or chi-square for categorical variables; Bold p-values indicate statistical significance

Comment 2: Suggest to add a flow chart of inclusion and exclusion criteria, which will be easier to understand.

Response: We agree that a flowchart would make the exclusion criteria for this secondary analysis easier to understand. A flowchart was added to this manuscript and is displayed below. The text related to analyses exclusion criteria under the Participant Characteristics subheading in the Method section was also reworded to reflect this addition to the manuscript and is below in bold, blue font.

Methods (Lines 146-147, 158-159):

“For this current analysis, participants were excluded based on the criteria described in Fig 1.”

“Figure 1. Flowchart of Exclusions from Analytic Sample. This flowchart reports the number and reasons for excluding participants for this secondary analysis.”

Comment 3: Result: The p value in the same table seems to be analyzed in different way (such as line 268), Please clarify blow the table

Response: A statement was added to the Table 1 description in the Results to clarify how the p-values were calculated in this table. This statement is also presented below.

Table 1 Description (Lines 456-457):

“p-values reflect a hypothesis test across groups using one-way ANOVA for continuous variables or chi-square for categorical variables;”

Comments from the reviewer 

Comment 1: Line 29, "and hours and OPA", is there a typo?

Response: Thank you for bringing this sentence from the Abstract to our attention. Employment status and average hours worked per week were reported in a demographic questionnaire, while OPA questions were reported in the Pregnancy Physical Activity Questionnaire. This sentence was reworded to better reflect how these variables were reported. All changes are displayed in the revised manuscript and below each comment in bold, blue font below.

Abstract (Line 29):

“Using employment status/job hours (self-reported in demographic questionnaires) and OPA from the Pregnancy Physical Activity Questionnaire, latent class analysis identified three groups:”

Comment 2: Line 40, suggest rephrasing "objective monitors"

Response: Based on this suggestion, we made an edit to line 40 of the Abstract section to rephrase the “objective monitors” statement to device-based activity as observed below. We have also made this change throughout the manuscript for clarity of reporting.

 Abstract (Line 40):

“All-day device-based activity differed across groups;”

Comment 3: Line 41, It is unclear what the p value in this sentence is for. There seems to be multiple comparisons.

Response: Thank you for noting the lack of additional p-values in this sentence from the Abstract. Adjustments were made to clarify line 41 and are below. 

Abstract (Lines 40-41):

“for example, the sitting group had the highest SB across trimester (all p<0.01) while the active group had the highest steps per day across trimester (all p<0.01).

Comment 4: Line 124 and Line 174, intraclass correlation coefficient was provided to indicate reproducibility. Isn't test-rest reliability a better term for reproducibility? Clarify if this was used in the original publication.

Response: We updated the manuscript to include the phrase “test-retest reliability” rather than reproducibility. These edits were made in the first paragraphs of the Leisure-time Physical Activity and Occupational Physical Activity sections of the Methods and are displayed below.

 Methods (Lines 164 and 198):

“The PPAQ is well validated for OPA with high test-retest reliability (intraclass correlation coefficient=0.93) and wide use in pregnant populations”

“Test-retest reliability of the sports/exercise questions from this questionnaire has been found to be good with an intra-class correlation coefficient of 0.83

“The PPAQ is well validated for OPA with high test-retest reliability (intra-class correlation = 0.93) and wide use in pregnant populations.(19)”

Comment 5: Line 139, clarify how proportions of work time spent in each category were calculated. The answer options in the questionnaire were in hours per day. What was the reason to convert to proportions?

Response: We agree that how proportions were calculated can be clarified within this manuscript. This clarification is provided below and in the Occupational Physical Activity section of the Methods in the manuscript. We converted questionnaire answers from hours per day to proportions due to varying work durations across participants (despite only daily estimates from the PPAQ). The participants in our study had both full-time and part-time work schedules, so we converted the hours per day (from the PPAQ) to proportions, then multiplied by the total number of work hours per week. 

 Methods (Lines 187-188):

“To calculate weekly durations of OPA in each category, daily proportions of work time spent sitting, standing and light walking, and walking quickly were calculated by dividing the time spent in each of these three OPA categories (reported ‘per day’) by the sum of all three OPA variables within each trimester.”

Comment 6: Line 143, when participants did not complete all visits, why were the averages of available data used, rather than the last available data, especially if the last visit was missing. Were missing data distributed evenly across the visits?

Response: Missing data were not evenly distributed across trimesters and were greatest in the third trimester due to losses to follow up and early deliveries. Our intention when averaging the OPA exposure across pregnancy was to understand associations with the accumulated exposure. For this reason, we decided it was most correct to weight all available data equally rather than overweight the most recent exposure to accommodate missing data at any trimester visit. Further, missing data was minimal, and we considered trimester specific associations. Thus, we expect that this methodological decision minimally impacts the results. 

Comment 7: Paragraph starting 146, have the authors considered moving this to the "Statistics" section? It talks about analysis.

Response: This is a great suggestion. We moved this last paragraph of the Occupational Physical Activity section of the Methods to be the first paragraph of the Statistical analyses section of the manuscript. This change is reflected in the manuscript to see this change.

Comment 8: Line 156, "in a mixture all three activities", clarify this phrase

Response: We agree that this phrase could be clarified, so we adjusted it in the Methods section within the first section of the Statistical Analyses section. 

 Methods (Line 386):

“The group that spent their workday in a mixture of OPA including sitting, standing and light walking, and walking quickly (i.e., the part-time mixed group; n=9) included a pre-school teacher, interpreter, and some healthcare workers (e.g., nurse practitioner and social worker).”

Comment 9: Line 175 and other places: what about putting the sample sizes in a table or figure? The way they are presented in separate paragraphs is confusing and hard to grasp.

Response: We agree that the trimester specific all-day activity sample sizes per trimester could be presented in a clearer manner. We added Supplemental Table 2 to condense these sample sizes to one location. The in-text mention of these sample sizes in the separate paragraphs of the Methods section were removed, and a statement was included in the first paragraph of the Device-based Activity Pattern Assessment section of the Methods and is below. 

Methods (Lines 207-208 and 1098-1100):

“Sample sizes for all-day activity variables by group and trimester are presented in Supplemental Table 2. Missing device-measured activity data was 7% in the first trimester, 12% in the second trimester, and 19% in the third trimester.”

Supplemental Table 2. Sample sizes for All-day Activity Data by Group as Presented in Table 2 

 Sitting Part-time Mixed Active Non-Working

SB, Prolonged SB, and Steps per day

1st Trimester 59 8 27 28

2nd Trimester 57 9 27 25

3rd Trimester 56 8 24 23

LPA

1st Trimester 58 7 26 27

2nd Trimester 57 7 27 24

3rd Trimester 54 7 23 21

MVPA 

1st Trimester 59 8 27 28

2nd Trimester 58 7 28 24

3rd Trimester 54 8 23 22

Data are presented as n values; Abbreviations: MVPA=moderate-to-vigorous physical activity, LPA=light physical activity, SB=sedentary behavior

Comment 10: Line 186, it seems sleep time was determined based on diaries. For those who did not report sleep times and days they did not report, how was it determined? What was the percentage of women reporting sleep in the diary on a daily basis?

Response: For those who did not report sleep times in the diary (approximately 4.7% of participants across the three visits), sleep times were determined from the activity monitors alone. This was clarified in the first paragraph of the Device-based activity pattern assessment section in the Methods and below.

 Methods (Lines 276-282): 

“Participants were asked to complete a diary to record if the device was removed and participant sleep onset and offset times. Data were exported and processed with PAL Technologies software (version 7.2.38; PAL Technologies, Glasgow, Scotland) and participant diary entries were used to remove any reported non-wear periods and sleep time using standardized methods.(22,23) In rare cases where participants did not report sleep times in the diary (4.7% of participants across the three visits), the device-based activity data were utilized to estimate sleep times.” 

Comment 11: Line 236 and other places, suggest be careful when using "objective activity". Light intensity PA was calculated using data from 2 monitors.

Response: We appreciate you mentioning this wording. Within the manuscript, we have changed the phrase “objective activity” or “objective monitors” to device-based activity or monitoring. These changes can be observed throughout the manuscript and below. 

 Abstract (Lines 32 and 40): 

“Device-based physical activity (ActiGraph GT3X), SB (activPAL3 micro), and blood pressure were measured each trimester.”

“All-day device-based activity differed across groups;”

Background (Line 127):

“We hypothesized that device-measured physical activity…”

Methods (Lines 205, 281, and 400): 

“All-day device-based physical activity was measured”

“the device-based activity data was utilized to estimate sleep times,”

“evaluated differences in device-based activity variables across groups”

Results (Lines 460 and 470):

“Table 2 presents device-based activity measures by group”

“Table 2. Device-based All-day Activity Quantification by Group”

Discussion (Lines 626, 684, and 704):

“Steeves et al. concluded that device-measured all-day physical activity levels in non-pregnant adults differed between low, intermediate, and high OPA groups”

“In our small MoM Health cohort (n=120) that made up the majority of the sample for the current secondary analysis, we found that being in the highest device-measured sedentary behavior group (~11 hours per day) was associated with higher blood pressure”

“Strengths of this secondary analysis include prospective OPA measurements using a validated questionnaire, the device-based measures of all-day activity,

Comment 12: Please report wear time of both monitors, number of days and number of hours worn per day.

Response: This is a great suggestion. Wear times were added as Supplemental Table 1 and are included below.

Supplemental Table 1. Average Wear Times for the activPAL and ActiGraph Monitors by Trimester

 Average Wear Days Average Wear Hours per Day

activPAL

Trimester 1 (n=122) 6.9 (0.9) 14.9 (1.0)

Trimester 2 (n=117) 6.9 (1.1) 15.1 (1.1)

Trimester 3 (n=111) 6.9 (1.0) 15.0 (1.0)

ActiGraph

Trimester 1 (n=122) 7.5 (1.4) 14.6 (1.2)

Trimester 2 (n=117) 7.2 (1.1) 14.5 (1.2)

Trimester 3 (n=107) 7.2 (0.9) 14.5 (1.5)

Data are presented as mean (sd)

Comment 13: Were days only one monitor was worn included in calculations? Was light intensity PA calculated using the same day of the 2 monitors or the averages of the few days each monitor was worn?

Response: Our protocol was for both monitors to be worn concurrently. Therefore, we used weekly averages of sedentary time and wear time from the activPAL monitor and MVPA from the ActiGraph to calculate LPA. We have clarified this in the Methods in the last paragraph under the Device-based Activity Pattern Assessment and below. 

 Methods (Lines 295-296):

“Average daily time spent in LPA was estimated using daily averages from the activPAL and GT3X as described above. LPA was calculated as follows: 100% of waking wear time minus [percent of time spent sedentary from activPAL + percent of time spent in MVPA from GT3X].

Comment 14: Table 2, clarify "% of wear time" is % of which monitor's wear time.

Response: Edits were made to better clarify which monitor was associated with the device-based activity variables in Table 2 and are presented below.

Table 2:

 Sitting 

(n=61) Part-time Mixed 

(n=9) Active 

(n=29) Non-Working

(n=32) p-value**

SB (% of activPAL wear time)

1st Trimester 67.9 (8.3) 63.1 (12.1) 59.1 (10.0) 65.5 (11.6) <0.001

2nd Trimester 65.4 (8.2) 67.0 (10.1) 58.1 (8.9) 62.4 (11.5) 0.004

3rd Trimester 65.3 (9.1) 65.7 (8.9) 58.7 (10.0) 62.6 (11.2) 0.007

Prolonged SB (≥30-minute bouts) (% of activPAL wear time)

1st Trimester 37.0 (11.6) 31.6 (15.5) 30.9 (9.6) 33.6 (13.9) 0.077

2nd Trimester 35.9 (11.0) 34.1 (12.0) 30.1 (11.4) 32.0 (12.0) 0.184

3rd Trimester 35.9 (10.9) 34.3 (11.2) 30.5 (10.6) 31.9 (14.1) 0.071

LPA (% of activPAL and ActiGraph wear time)

1st Trimester 28.3 (7.8) 32.0 (12.3) 36.8 (9.5) 31.8 (11.1) <0.001

2nd Trimester 30.8 (8.0) 30.0 (10.7) 37.7 (8.8) 35.1 (11.5) 0.008

3rd Trimester 31.8 (8.8) 32.1 (10.5) 38.1 (9.6) 35.0 (11.5) 0.021

MVPA (% of ActiGraph wear time)

1st Trimester 3.8 (1.9) 3.1 (1.2) 4.1 (1.7) 2.9 (1.9) 0.135

2nd Trimester 3.8 (1.9) 3.8 (1.7) 4.4 (1.8) 3.3 (1.6) 0.034

3rd Trimester 3.0 (1.9) 2.8 (1.4) 3.3 (1.6) 2.9 (1.7) 0.602

Steps per day

1st Trimester 7,274 (2,591) 7,594 (2,239) 8,891 (2,582) 6,587 (2,456) <0.001

2nd Trimester 7,483 (2,697) 6,380 (2,075) 8,899 (2,811) 7,006 (2,312) 0.003

3rd Trimester 6,608 (2,298) 6,439 (1,276) 8,167 (2,160) 6,867 (2,816) 0.002

Comment 15: Line 268 and other places, are the p values in the table for the overall ANOVA test? Were follow-up tests done for pairwise comparison? The sentence seems to about comparing MVPA in the active group compared to other groups, which is different from the overall ANOVA test. Similar interpretations were made in several other places.

Response: We made changes to the manuscript for a clearer interpretation of the p-values from Table 2 which came from omnibus tests (likelihood ratio tests) conducted within nested linear regression models. Per the reviewer’s suggestion, additional post-hoc testing was conducted. Changes were made to the Statistical Analyses section of the Methods and under the Activity Pattern Results section. These changes are below.

Statistical analyses (Lines 409-412):

“When significant differences were detected, pairwise post hoc analyses with Bonferroni adjustment for multiple comparisons were conducted to determine the presence of statistical difference between groups in all-day activity”

Results (Lines 461-467): 

“SB was significantly different across groups in each trimester (all p<0.05) with the active group having lower SB compared to the sitting group (post hoc p<0.05). LPA also differed across groups in each trimester (all p<0.03), with the active group having higher LPA compared to the sitting group (post hoc p<0.05). Steps per day differed across group in each trimester (all p<0.004), with the active group having higher steps per day compared to the other three groups (post hoc p<0.05). MVPA differed by group only in the second trimester (p=0.034), though pairwise differences were not statistically different after adjustment for multiple comparisons. No differences were observed across groups for MVPA during the first and third trimester or prolonged SB in any trimester (all p>0.07).”

Table 2. Device-based All-day Activity Quantification by Group

 Sitting 

(n=61) Part-time Mixed 

(n=9) Active 

(n=29) Non-Working

(n=32) p-value**

SB (% of activPAL wear time)

1st Trimestera 67.9 (8.3) 63.1 (12.1) 59.1 (10.0) 65.5 (11.6) <0.001

2nd Trimestera 65.4 (8.2) 67.0 (10.1) 58.1 (8.9) 62.4 (11.5) 0.004

3rd Trimestera 65.3 (9.1) 65.7 (8.9) 58.7 (10.0) 62.6 (11.2) 0.007

Prolonged SB (≥30-minute bouts) (% of activPAL wear time)

1st Trimester 37.0 (11.6) 31.6 (15.5) 30.9 (9.6) 33.6 (13.9) 0.077

2nd Trimester 35.9 (11.0) 34.1 (12.0) 30.1 (11.4) 32.0 (12.0) 0.184

3rd Trimester 35.9 (10.9) 34.3 (11.2) 30.5 (10.6) 31.9 (14.1) 0.071

LPA (% of activPAL and ActiGraph wear time)

1st Trimestera 28.3 (7.8) 32.0 (12.3) 36.8 (9.5) 31.8 (11.1) <0.001

2nd Trimestera 30.8 (8.0) 30.0 (10.7) 37.7 (8.8) 35.1 (11.5) 0.008

3rd Trimestera 31.8 (8.8) 32.1 (10.5) 38.1 (9.6) 35.0 (11.5) 0.021

MVPA (% of ActiGraph wear time)

1st Trimester 3.8 (1.9) 3.1 (1.2) 4.1 (1.7) 2.9 (1.9) 0.135

2nd Trimester 3.8 (1.9) 3.8 (1.7) 4.4 (1.8) 3.3 (1.6) 0.034

3rd Trimester 3.0 (1.9) 2.8 (1.4) 3.3 (1.6) 2.9 (1.7) 0.602

Steps per day

1st Trimesterb 7,274 (2,591) 7,594 (2,239) 8,891 (2,582) 6,587 (2,456) <0.001

2nd Trimesterb,c 7,483 (2,697) 6,380 (2,075) 8,899 (2,811) 7,006 (2,312) 0.003

3rd Trimestera,c 6,608 (2,298) 6,439 (1,276) 8,167 (2,160) 6,867 (2,816) 0.002

Data are presented as mean (SD); n values represent maximum sample size per group; 

** Likelihood ratio test; Adjusted for age, pre-pregnancy BMI, race, and education level; 

Superscript letters correspond to pairwise differences observed in post hoc testing with Bonferroni adjustment for multiple comparisons: a=significant difference between sitting versus active group, b=significant difference between non-working versus active group, c= significant difference between mixed versus active group

Abbreviations: MVPA=moderate-to-vigorous physical activity, LPA=light physical activity, SB=sedentary behavior, OPA=occupational physical activity; Bold p-values indicate statistical significance

Comment 16: Were leisure time PA reported in Results, since it was talked about in Methods.

Response: Leisure-time physical activity was reported in the Results at the bottom of Table 1 and within the last sentence of the first paragraph in the Results section. These are both included below and highlighted in yellow for your reference. 

Results (Lines 442-443):

“Pre-pregnancy BMI, race, and self-reported leisure-time physical activity did not differ between groups.”

Table 1. Participant Characteristics by Group

 Total

(n=131) Sitting

(n=61) Part-time Mixed

(n=9) Active

(n=29) Non-working

(n=32) p-value

Age (years) 30.9 ± 4.9 32.2 ± 4.1 31.4 ± 4.0 30.6 ± 5.4 28.3 ± 5.1 0.003

Pre-Pregnancy BMI (kg/m2) 26.8 ± 6.7 26.3 ± 5.9 24.1 ± 3.9 29.1 ± 8.0 26.5 ± 7.3 0.158

Race 0.166

White 101 (77.1%) 53 (86.9%) 7 (77.8%) 21 (72.4%) 20 (62.5%) 

Black 23 (17.6%) 5 (8.2%) 2 (22.2%) 6 (20.7%) 10 (31.2%) 

Other 7 (5.3%) 3 (4.9%) 0 (0.0%) 2 (6.9%) 2 (6.2%) 

Education <0.001

Less than a Bachelor’s Degree 44 

(33.6%) 9 

(14.8%) 2 

(22.2%) 13 

(44.8%) 20 

(62.5%) 

Bachelor’s Degree or Higher 87 

(66.4%) 52 

(85.2%) 7 

(77.8%) 16 

(55.2%) 12 

(37.5%) 

Self-Reported Leisure-time PA (MET-h/week)

1st Trimester (n=130) 17.0 

(14.9) 18.0 

(15.2) 22.1 

(18.3) 15.0 

(12.0) 15.6 

(14.9) 0.535

2nd Trimester (n=121) 15.3 

(13.9) 15.1 

(14.4) 21.2 

(18.0) 14.0 

(11.9) 15.3

(13.5) 0.647

3rd Trimester (n=117) 11.9 

(13.4) 12.5 

(14.2) 13.6 

(13.7) 8.4 

(10.0) 13.4 

(14.9) 0.521

Data presented as n (%) or mean (SD); Abbreviations: BMI=body mass index, SD= standard deviation, PA=physical activity, MET=metabolic equivalents, OPA=occupational physical activity; p-values reflect a hypothesis test across groups using one-way ANOVA for continuous variables or chi-square for categorical variables; Bold p-values indicate statistical significance 

Comment 17: Why was study site not adjusted?

Response: We added Supplemental Table 1 (above in a comment to the editor) that includes study site differences which were minimal. We also added a statement regarding our rationale for not adjusting for site in the second paragraph of the Statistical Analyses section and below.

Methods (Lines 416-419):

“Study site was also considered as a covariate but was ultimately not included due to minimal differences in participant characteristics between sites, concerns of parsimony due to the small sample, and minimal differences in sensitivity analyses that included site.”

Comment 18: It mentions that the "mixed actvities" group had "part-time" work hours. It is not clear this was actually included in Results. Please clarify. Perhaps provide number of hours worked for each group? Especially this was not adjusted in the models.

Response: Work hours per group were reported in Figure 1 as the job hours near the top of each bar graph. This figure is included below for your reference. In the second paragraph of the Statistical Analyses section, we have a statement that describes this difference and mentions this figure. To add clarity, we now report the average hours worked per week to this sentence (see below). We have this information in the Methods and not the Results reflecting that we considered job hours as a covariate during model building, but ultimately had to exclude it due to collinearity (below for your reference). 

Statistical analyses (Lines 381-382, 414-416):

“Average hours per week spent in the three OPA intensities differed by latent class groups (p<0.001) with the part-time mixed group working less hours (14.6 hours per week) compared to the sitting (40.4 hours per week) and active groups (40.0 hours per week) as presented in Fig 2.”

“Job hours were considered as a covariate but were ultimately not included due to collinearity with the group variables that resulted in variance inflation factors greater than 10”

Comment 19: Line 376, was "leisure-time SB" correct in this sentence?

Response: We updated the second to last paragraph of the discussion section to improve clarity as follows.

 Discussion (Line 690):

“Taken together, specific research on the associations of overall and occupational SB with pregnancy and infant health outcomes is an area in need of more research.”

Comment 20: Need to discuss limitations of using 2 monitors. Would any category of time spent be under/overestimated?

Response: A sentence was added in the last paragraph of the Discussion regarding the use of two activity monitors.

Discussion (Lines 705-708):

“Although the device-based monitors used for this study are best practice measures to determine SB and MVPA, (45,46) LPA was calculated based on the two devices rather than directly measured, which may have introduced additional error for the measurement of LPA.”

---

## [Decision Letter · Decision Letter 1]

10 Dec 2023

Occupational physical activity as a determinant of daytime activity patterns and pregnancy and infant health

PONE-D-23-20019R1

Dear Dr. Thrower,

We’re pleased to inform you that your manuscript has been judged scientifically suitable for publication and will be formally accepted for publication once it meets all outstanding technical requirements.

Kind regards,

Zhaoxia Liang

Academic Editor

PLOS ONE

Additional Editor Comments (optional):

All comments have been addressed.

Reviewers' comments:

Reviewer's Responses to Questions

**Comments to the Author**

1. If the authors have adequately addressed your comments raised in a previous round of review and you feel that this manuscript is now acceptable for publication, you may indicate that here to bypass the “Comments to the Author” section, enter your conflict of interest statement in the “Confidential to Editor” section, and submit your "Accept" recommendation.

Reviewer #1: All comments have been addressed

2. Is the manuscript technically sound, and do the data support the conclusions?

Reviewer #1: Yes

3. Has the statistical analysis been performed appropriately and rigorously? 

Reviewer #1: Yes

4. Have the authors made all data underlying the findings in their manuscript fully available?

Reviewer #1: Yes

5. Is the manuscript presented in an intelligible fashion and written in standard English?

Reviewer #1: Yes

6. Review Comments to the Author

Reviewer #1: (No Response)

7. PLOS authors have the option to publish the peer review history of their article (what does this mean?). If published, this will include your full peer review and any attached files.

Reviewer #1: No

---

## [Editor Report · Acceptance letter]

13 Dec 2023

PONE-D-23-20019R1 

PLOS ONE

Dear Dr. Thrower, 

I'm pleased to inform you that your manuscript has been deemed suitable for publication in PLOS ONE. Congratulations! Your manuscript is now being handed over to our production team.

Kind regards, 

on behalf of

Dr. Zhaoxia Liang 

Academic Editor

PLOS ONE